# Incidence of cerebrovascular disease in Peru from 2015 to 2023

**Diego Maximiliano Guevara Rodríguez**[ID]*°, **Juan Diego Pichihua Grandez**[ID]°, **Fabrizio Valdivia Dianderas**[ID]°, **José Del Carmen Sara**[ID]

Faculty of Human Medicine, Universidad de Piura, Lima, Peru

☙ These authors contributed equally to this work.
* diego.guevara@alum.udep.edu.pe

## Abstract

Cerebrovascular disease (stroke) is one of the leading causes of mortality and disability worldwide, particularly in low- and middle-income countries. This study aims to estimate the incidence of stroke in Peru between 2015 and 2023 using national hospital discharge data provided by the National Health Superintendency. We conducted a mixed ecological study using records of stroke cases reported across various healthcare systems, including the Ministry of Health, Social Security, and private entities. Hospitalizations were categorized according to ICD-10 codes (I60-I64) and stratified by age, sex, and region. Incidence rates were calculated using population projections from the National Institute of Statistics and Informatics. A total of 89,776 hospital discharges for stroke were analyzed, yielding an incidence rate of 3.11 per 10,000 persons over the study period, with a predominance in men and individuals over 60 years of age. Cerebral infarction was the most common diagnosis, particularly among those over 40 years old. Incidence varied significantly across regions, with Lima and Callao consistently exceeding the national average. The results highlight disparities in healthcare access and the need for targeted public health interventions. Our findings provide a 9-year overview of stroke in Peru, offering evidence to estimate hospital bed demand and prioritize preventive and management strategies—particularly in regions with higher vulnerability.

## Author summary

Cerebrovascular disease comprises a spectrum of diseases, from strokes to hemorrhages of the nervous system. This pathology is one of the leading causes of disability and death worldwide. In addition, there is an increase in the incidence of cerebrovascular disease worldwide, both in rich and middle-income countries, such as Peru. We found that in Peru there are 3 cases of cerebrovascular disease per 10,000 people each year, and it was more frequent in older

provided the original author and source are credited.

**Data availability statement:** **AT ACCEPT: Ask the authors to update their data availability*** We will publish the raw data once the manuscript is accepted.

**Funding:** The author(s) received no specific funding for this work.

**Competing interests:** The authors have declared that no competing interests exist.

adults and males. The capital, Lima, together with Callao were the regions that exceeded 3 cases per 10,000 people during the study period. These jurisdictions concentrate the largest number of hospitals and specialized professionals, which reveals inequalities in the distribution of health resources. These findings allow recommendations to be made to improve diagnostic capacity through imaging technologies in the different regions of the country, in addition to strengthening the referral system of primary care facilities for more specific and specialized management.

## Introduction

Cerebrovascular disease (stroke) encompasses a range of disorders caused by vascular damage in the brain or spinal cord, usually with an acute onset. Stroke includes ischemic events, caused by hypoperfusion secondary to atherosclerotic stenosis or embolism, and hemorrhagic events, such as intracerebral or subarachnoid hemorrhage [1].

Globally, stroke is the second leading cause of cardiovascular mortality, with a constant rise in its prevalence. According to the Global Burden of Disease (GBD) 2021, global prevalence increased from 0.97% in 1990 to 1.23% in 2021, and incidence rose from 13.2 to 15.1 cases per 10,000 persons over the same period [2]. The cumulative incidence of stroke varies significantly across countries. Between 1989 and 2001, Sweden reported an incidence of 6.38 per 10,000 persons [3]. In South America, incidence rates in Chile and Ecuador ranged between 2.74 and 6.5 cases per 10,000 persons, depending on gender and the study period [4,5]. In Peru, by 2021, the incidence ranged between 6.1 and 6.8 cases per 10,000 persons [2].

These differences highlight not only demographic and healthcare access variations but also potential inequalities in the distribution of risk factors, such as hypertension and obesity. It is estimated that up to 91% of ischemic and 87% of hemorrhagic events occur in people with modifiable risk factors; therefore, primary prevention of stroke is vital for its prevention and timely management [6]. The increase in stroke frequency suggests potential deficiencies in the primary prevention of its causes [7] and may indicate an increase in the prevalence of related non-communicable diseases and overloading of health services, especially the inpatient care unit.

In Peru, epidemiological information on stroke is limited. A study conducted in Cusco, a high-Andean city, reported a prevalence of 6.47 cases per 1,000 persons in 1988 [8], and another study using hospital records from 2017 and 2018 estimated a cumulative incidence of 3.32 and 3.99 per 10,000 person-years, respectively [9]. However, no studies have estimated the incidence of stroke over time and its geographic distribution, which is essential for better resource planning.

Studies using hospital discharge data for other diseases, such as Guillain-Barré syndrome [10] and respiratory syncytial virus [11], support the use of national hospital discharge databases to provide valuable information on disease burden and optimize healthcare resource planning. This methodology is applicable to stroke, allowing for

the evaluation of its trends over time and its distribution across subgroups, which helps identify healthcare access inequalities and in sizing the demand for hospital resources to care for this disease. In Peru, the National Superintendence of Health (SUSALUD) has collected nationwide hospital discharge data since 2015 [12], enabling the quantification and characterization of stroke, whose management typically requires hospitalization to ensure favorable outcomes [13].

The aim of this study was to estimate the incidence of stroke between 2015 and 2023 in Peru, and in subgroups defined by age, sex, and geographic location. We hypothesize a progressive increase in incidence during this period. These results could offer valuable insights to guide decision-making in the allocation of health sector resources, both to address the demand generated by stroke across Peru's departments and to identify high-risk population subgroups, enabling a focused approach to primary prevention and health education actions.

## Methods

### Study setting

Peru is a South American country divided into 24 departments, subdivided into 196 provinces [14]. Lima, the country's capital, and Callao have the highest concentration of healthcare facilities [15].

The Peruvian healthcare system has a mixed structure, divided into two sectors: public and private. The public sector consists of several subsystems: the Social Security (SS), which serves formal workers and their families, and the health insurance from the Ministry of Health (SIS-MINSA), which provides subsidized services to the uninsured population. The Ministry of Health manages the network of hospitals and specialized institutes, mainly in Lima, while the regional and local governments administer the rest of the hospitals, health centers, and clinics. Additionally, the Armed Forces and the National Police of Peru have their own health systems. In the private sector, both profit-driven entities, such as health service providers (EPS) and private insurers, coexist with non-profit organizations funded by various sources [16,17].

The regulatory entity overseeing all health systems is the National Superintendence of Health (SUSALUD), which ensures the proper functioning of healthcare services, including insurance coverage and institutional quality [18].

### Study design

We conducted a mixed ecological study using a database of hospital discharge diagnoses in Peru. SUSALUD creates this database from the formal records of each healthcare institution (IPRESS), which submit monthly reports with year, month, age, sex, ICD-10 codes, and the number of cases treated. After verifying and correcting the records, SUSALUD validates the information and corrects errors. Each column in the database represents a variable, and the last column records the total number of hospital discharges. We filtered cerebrovascular disease cases using ICD-10 codes ranging from I60.0 to I64.X (Fig 1). Additionally, we used population projections by calendar year for age and department from the National Institute of Statistics and Informatics (INEI) published in 2019 and 2020, respectively [19,20].

The database includes hospital discharges from the Ministry of Health (MoH), Social Security (SS), Regional Health Directorates, Armed Forces and Police Health Services, and private entities. These health establishments are classified in three categories of care according to their resolution capacity, equipment, and technology. The categories, in ascending order of complexity, are I-1, I-2, I-3, I-4, II-1, II-2, II-E, III-1, III-2, III-E.

The first level of complexity (category I) offers basic services, health promotion and disease prevention; at this level, medical posts are classified as category I-1, where there are often no permanent physicians due to lack of resources or their remote location, and where nursing technicians or other health professionals are in charge. Category I-2 includes health centers or clinics that generally serve small populations and provide basic outpatient care, outpatient consultation, promotion, prevention and control of common diseases, but do not perform complex procedures or deliveries. Category I-3 includes health centers or polyclinics that have general practitioners, pediatricians, obstetricians, and nursing personnel, and where basic procedures such as cures and minor sutures are performed, but no hospitalization is available.

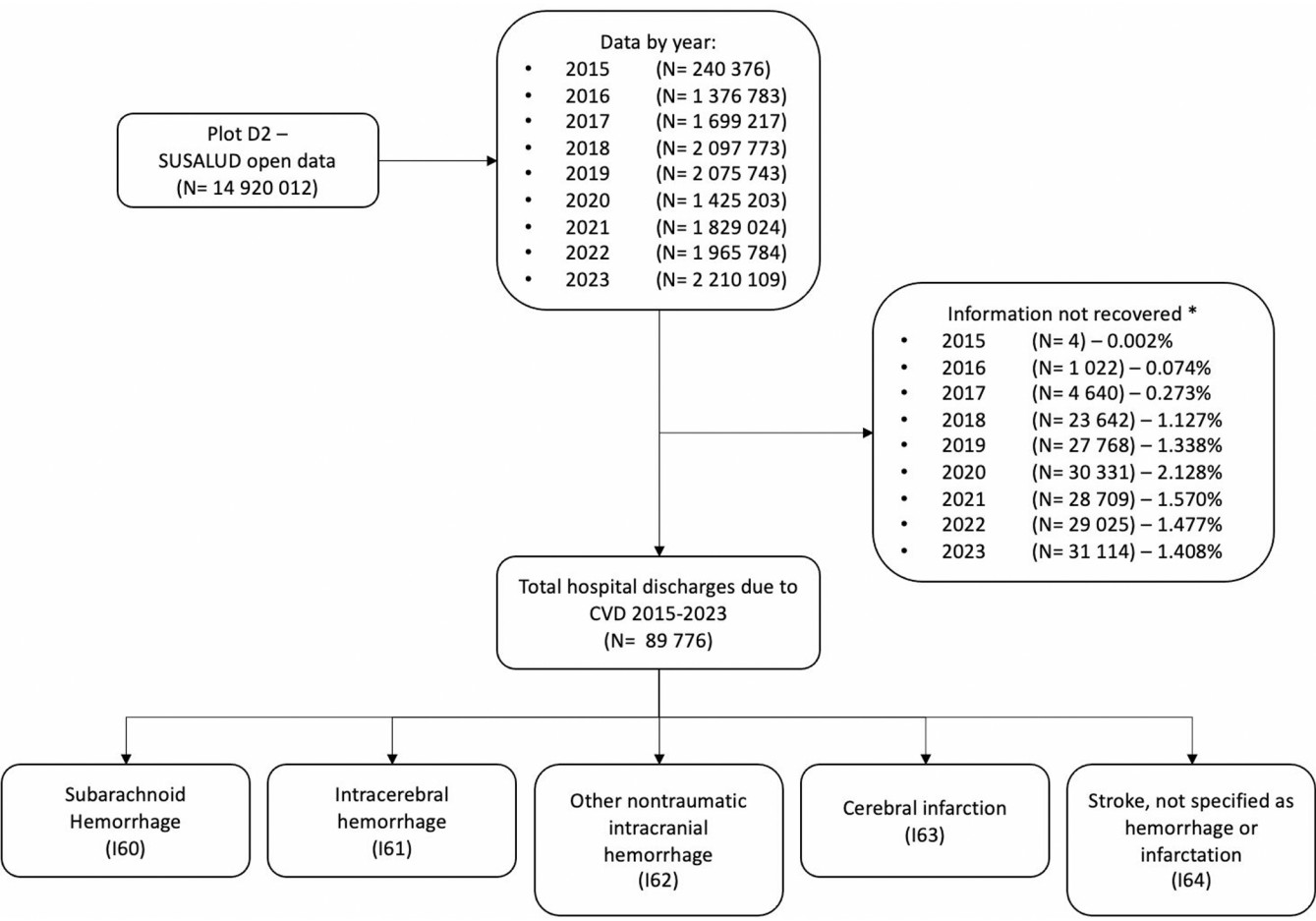

**Fig 1. List of selected codes.**

Category I-4 includes medical centers, maternal and child centers where there are differentiated areas for outpatient consultation, minor procedures, and care for simple emergencies and childbirth, with spaces for a short stay for postpartum hospitalization or recovery. [21,22].

The second level of complexity (category II) includes hospitals that offer diagnosis and treatment of medium complexity diseases, such as general hospitals, provincial hospitals that have basic specialties, such as pediatrics, gynecology, general surgery, internal medicine, among others, and have the capacity for hospitalization, in addition to performing minor or intermediate surgeries. In category II-1, these facilities have outpatient, emergency and hospitalization services, while in category II-2, intensive care units are added. Category II-E includes facilities that provide health services in a main specialty, and may even have subspecialized services with respect to the main specialty [21,22].

The third level of complexity (category III) are hospitals and centers specializing in the treatment of complex diseases. Category III-1 includes facilities with specialist physicians for all specialties. In category III-E are those establishments that develop in a specialty, together with subspecialties on a mandatory basis. In category III-2 are specialized Health Institutes where health care is provided with the highest resolution capacity in a clinical field or age group. It also develops technological innovation, research and teaching [21,22].

## Variable definitions

The primary variable of interest was the number of hospital discharges due to stroke, defined by cases categorized under subarachnoid hemorrhage (I60), intracerebral hemorrhage (I61), non-traumatic subdural hemorrhage (I62), cerebral infarction (I63), and acute stroke not specified as hemorrhagic or ischemic (I64) from chapter IX of ICD-10 (Fig 1). The codes were selected based on previous research [3,5,9,23] and validated by a panel of experts, including a neurologist and an epidemiologist, to reduce measurement bias. We defined unspecified stroke diagnoses using codes I60.9, I61.9, I62.9, I63.9, I64.X, and classified the rest of the codes as specified diagnoses. This variable was calculated based on the total number of treated cases, which corresponded to the absolute frequency of hospital discharges. A hospital discharge occurs when a patient leaves the hospital after receiving care, either by discharge, transfer, or death [24].

We characterized the cases using variables such as age, grouped into seven age groups ranging from 0-9 years to 60 years and older; the year in which the hospital discharge was reported; and the complexity level of the healthcare institution (IPRESS), classified into categories 0 (unassigned), I, II, and III. We also considered the healthcare sector, grouping the IPRESS by the population they attended, including MoH, Armed Forces and Police Health Services (AF&PHS), Social Security (SS), and Regional Health Directorates (local governments and municipalities).

## Data processing

We used Excel 2020 to extract data with ICD-10 codes related to stroke and create a new database. Rows that lacked information on ICD-10 diagnostic codes, the total number of cases treated, and sex were considered missing data (Fig 1). The generated database included the following variables in each column, in the following order: year of report, month of report, healthcare sector, complexity level of the healthcare establishment, sex, age, ICD-10 diagnostic code category and subcategory, ICD-10 diagnostic code category only, specificity of the diagnosis, and the total number of cases treated (S1 Table).

## Data analysis

We used Excel pivot tables to analyze the frequency of hospital discharges due to stroke by year, age, sex, department, diagnostic category, complexity level of the healthcare institution, and healthcare sector. To estimate the incidence, we use hospital discharges due to stroke as the numerator, and - as the population at risk, the denominator - we use the population projections by years for sex, age and departments of the INEI [19,20]. The cumulative incidence was expressed per 10,000. We estimated the cumulative incidence for each year of the study and its trends across age groups, sex, departments, and diagnostic categories. Additionally, we determined the proportion of specified diagnoses by year, age, sex, IPRESS category, and healthcare sector. We also displayed the most frequent specified diagnoses in each age group. Graphs were created using GraphPad Prism version 10.2.0.

## Ethical statements

As this study used secondary data, it did not involve interaction with human participants. The hospital discharge data were obtained from an open-access database from the National Superintendence of Health. The database was anonymized and publicly available. No researchers reported conflicts of interest. The research protocol was approved by the Institutional Ethics Committee of the University of Piura.

## Results

### Participant selection and characteristics

We evaluated 14,920,012 data records, of which 89,776 were hospital discharges due to stroke. Of these, 54.7% were male, and 71.2% were over 60 years old. The year 2018 accounted for the highest number of strokes hospital discharges

(15.3%). Regarding diagnostic categories, cerebral infarction was the most frequent, accounting for 40.3% of cases. Hospitals classified as level II complexity concentrated 51.1% of hospital discharges due to stroke, while healthcare institutions managed by regional governments accounted for 33.3% (Table 1) (S2 Table).

**Table 1. Frequency of hospital discharges due to cerebrovascular disease.**

| Variable | Hospital Discharges | |
|---|---|---|
| | N | % |
| **Year** | | |
| 2015 | 2721 | 3.0 |
| 2016 | 9422 | 10.5 |
| 2017 | 11352 | 12.6 |
| 2018 | 13758 | 15.3 |
| 2019 | 12814 | 14.3 |
| 2020 | 7897 | 8.8 |
| 2021 | 8946 | 10.0 |
| 2022 | 10783 | 12.0 |
| 2023 | 12083 | 13.5 |
| **Age group (y)** | | |
| 0–9 | 1342 | 1.5 |
| 10–19 | 1381 | 1.5 |
| 20–29 | 2095 | 2.3 |
| 30–39 | 3183 | 3.5 |
| 40–49 | 6239 | 6.9 |
| 50–59 | 11660 | 13.0 |
| 60+ | 63876 | 71.2 |
| **Gender** | | |
| Male | 49071 | 54.7 |
| Female | 40705 | 45.3 |
| **Sector of health services** | | |
| Regional Government | 29900 | 33.3 |
| Social Security | 28149 | 31.4 |
| Private health services | 14466 | 16.1 |
| Health Ministry | 13934 | 15.5 |
| Armed Forces and Police Health | 1921 | 2.1 |
| District Municipality | 1406 | 1.6 |
| **Category by complexity** | | |
| 0 *(Category not assigned)* | 1349 | 1.5 |
| I | 887 | 1.0 |
| II | 45861 | 51.1 |
| III | 41679 | 46.4 |
| **Diagnostic Categories** | | |
| Subarachnoid haemorrhage | 9386 | 10.5 |
| Intracerebral haemorrhage | 14033 | 15.6 |
| Other nontraumatic intracranial haemorrhage | 5560 | 6.2 |
| Cerebral infarction | 36137 | 40.3 |
| Stroke, not specified as haemorrhage or infarction | 24660 | 27.5 |
| **Total** | **89776** | **100** |

## Incidence of stroke by year and gender

The national average incidence for the 2015–2023 period was 3.11 cases of stroke per 10,000 persons. The year 2018 had the highest incidence, with 4.36 per 10,000 persons, while 2015 showed the lowest incidence with 0.91 cases per 10,000 persons. There were two periods of increasing trends: 2015–2018 and 2020–2023, with males being the most affected group in both periods (Fig 2).

## Incidence of stroke by department

The departments of Callao and Lima exceeded the national average in seven of the years studied. Additionally, other departments, such as Amazonas, Arequipa, Lambayeque, La Libertad, Ica, San Martín, Cusco, Moquegua, Tacna, Cajamarca, Junín, Ayacucho, Loreto, Madre de Dios, and Apurímac, surpassed the national average in specific years. Pasco recorded the highest incidence of stroke with 14.7 per 10,000 persons in 2020 (Fig 3).

## Incidence of stroke by age

The 60+ age group consistently showed the highest incidence of stroke from 2015 to 2023. The peak cumulative incidence occurred in 2018, with 26.7 cases per 10,000 persons. The second most affected group was individuals aged 50–59. People under 29 years of age had the lowest incidence of stroke, with less than 1 case per 10,000 persons, compared to those over 60 (Fig 4).

## Incidence of stroke by diagnostic category

Cerebral infarction (ICD code I63) had the highest incidence, peaking at 1.64 and 1.65 cases per 10,000 persons in 2018 and 2023, respectively. Non-traumatic subdural hemorrhage (I62) had the lowest incidence of stroke, with a peak of 0.29 per 10,000 persons in 2018 (Fig 5).

## Specified diagnosis frequency

The year 2015 had the highest proportion (42.78%) of specified diagnoses. In the 0–9 years age group, 55.37% of cases had a specified diagnosis. Differences by sex were minimal, with 34.25% of specified diagnoses in males and 34.70% in females (Table 2) (S3 Table). The Armed Forces and Police Health Services had the highest proportion of specified diagnoses, reaching 41.44%. Additionally, the complexity level of healthcare institutions influenced the proportion of specified diagnoses, with level III institutions showing a proportion of 43% (Fig 6).

## Stroke diagnosis by age group

Hemorrhagic categories of stroke were the most common diagnoses in individuals younger than 39 years. In infants under 1-year, intracerebral hemorrhage predominated, accounting for 18.2% of discharges. Among those aged 1–39 years, intracerebral and subdural hemorrhages were the most frequent diagnoses. From age 40 onwards, cerebral infarction became the leading diagnostic category, with cerebral artery embolism being the most frequent etiology (Table 3).

## Discussion

In our study, the average incidence was 3.11 per 10,000 persons, with men over 60 years being the most affected, particularly in Lima and Callao. According to GBD 2021 estimates for Peru, the cumulative incidence of stroke between 2015 and 2021 ranged from 6.4 to 6.8 per 10,000 persons. [2]. However, our results are approximately half of these estimates, which could be due to GBD's use of mathematical models that combine data from multiple sources, such as population

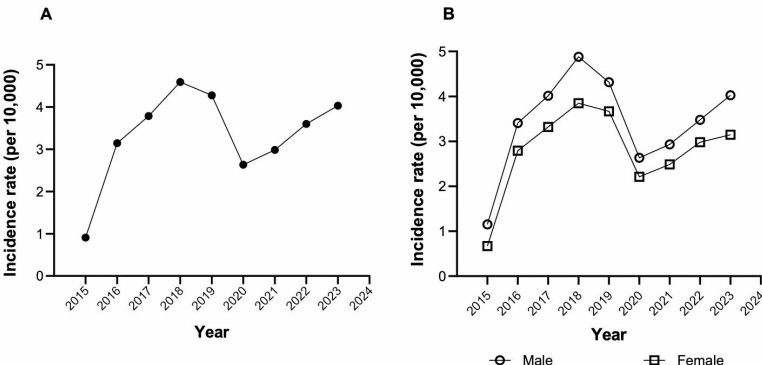

**Fig 2. Incidence of cerebrovascular disease by year and gender.**

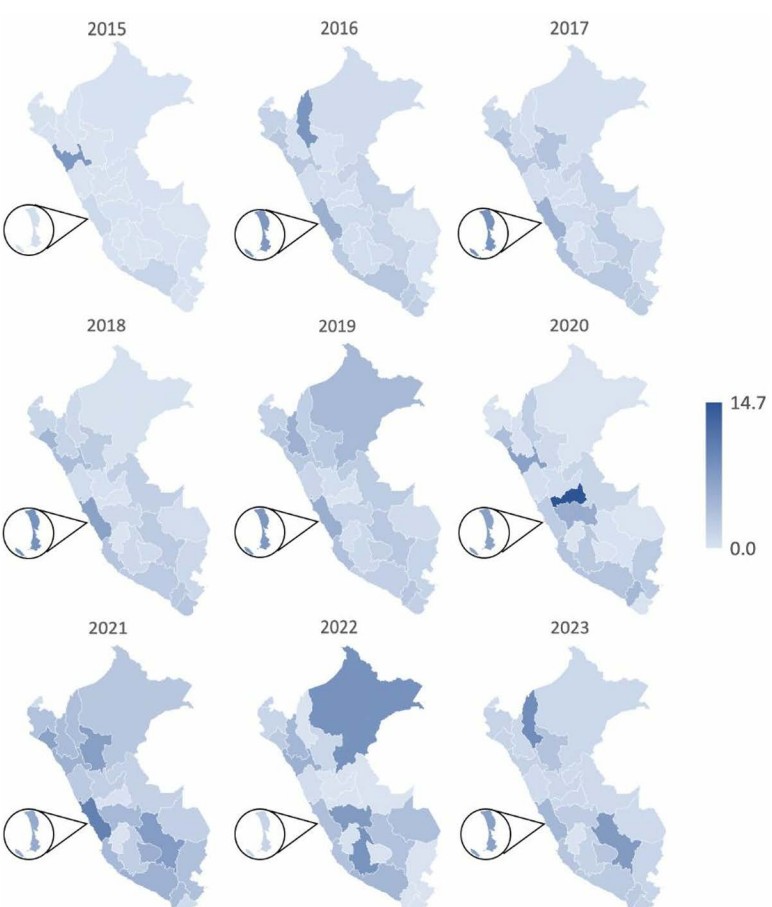

**Fig 3. Incidence of cerebrovascular disease by year and region per 10,000 persons.** The map was created with Excel. This program uses the base layer of the map from OpenStreetMap: https://www.openstreetmap.org/relation/288247.

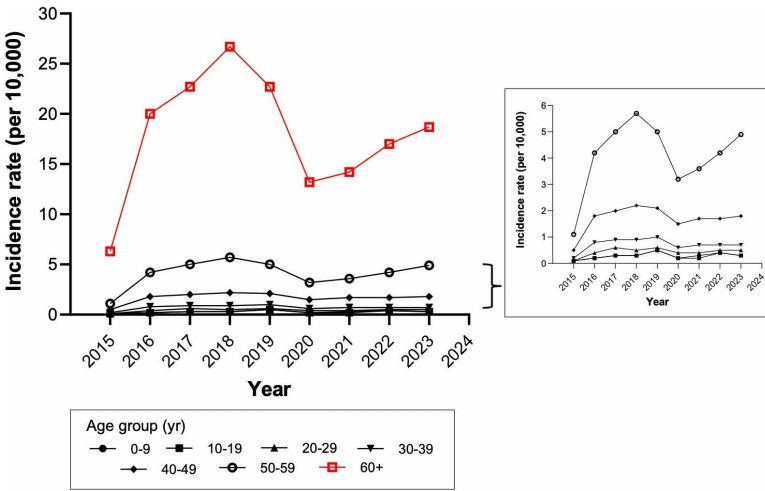

**Fig 4. Incidence of cerebrovascular disease by year and age group.**

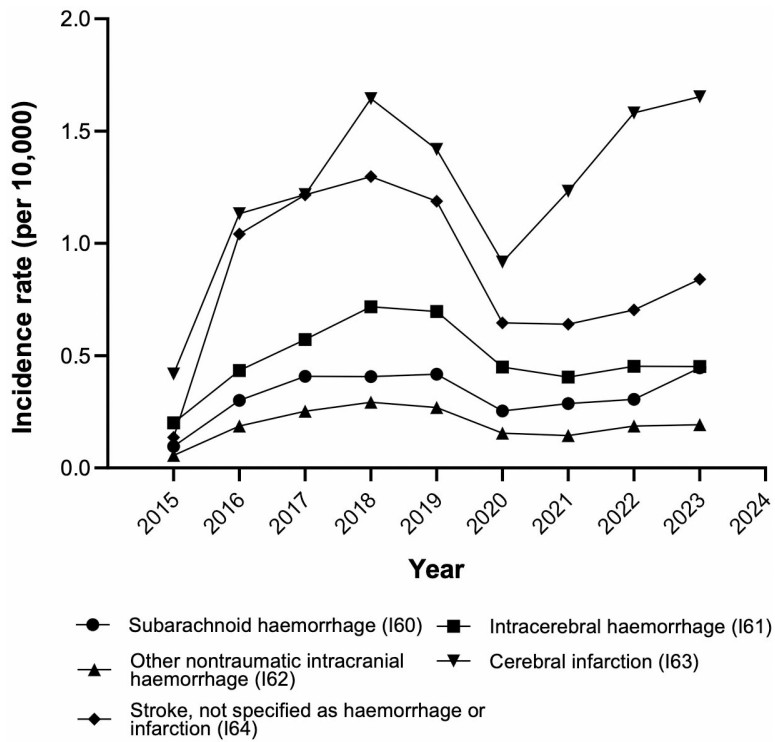

**Fig 5. Incidence of cerebrovascular disease according to year and diagnostic category.**

surveys and non-hospital records, potentially capturing cases not diagnosed in hospitals [25]. In contexts with barriers to healthcare access, these models may overestimate incidence. Additionally, GBD adjusts its estimates to account for comorbidities and other risk factors [25].

**Table 2. Specified diagnosis frequency by year, age group and gender.**

| Variable | Total (n) | Specified Diagnosis (%) |
|---|---|---|
| **Year** | | |
| 2015 | 2721 | 42.78 |
| 2016 | 9422 | 31.37 |
| 2017 | 11352 | 31.13 |
| 2018 | 13758 | 34.05 |
| 2019 | 12814 | 35.05 |
| 2020 | 7897 | 33.94 |
| 2021 | 8946 | 38.27 |
| 2022 | 10783 | 37.53 |
| 2023 | 12083 | 32.71 |
| **Age group (y)** | | |
| 0–9 | 1342 | 55.37 |
| 10–19 | 1381 | 46.92 |
| 20–29 | 2095 | 43.29 |
| 30–39 | 3183 | 42.38 |
| 40–49 | 6239 | 39.38 |
| 50–59 | 11660 | 35.70 |
| 60+ | 63876 | 32.35 |
| **Gender** | | |
| Male | 49071 | 34.25 |
| Female | 40705 | 34.70 |

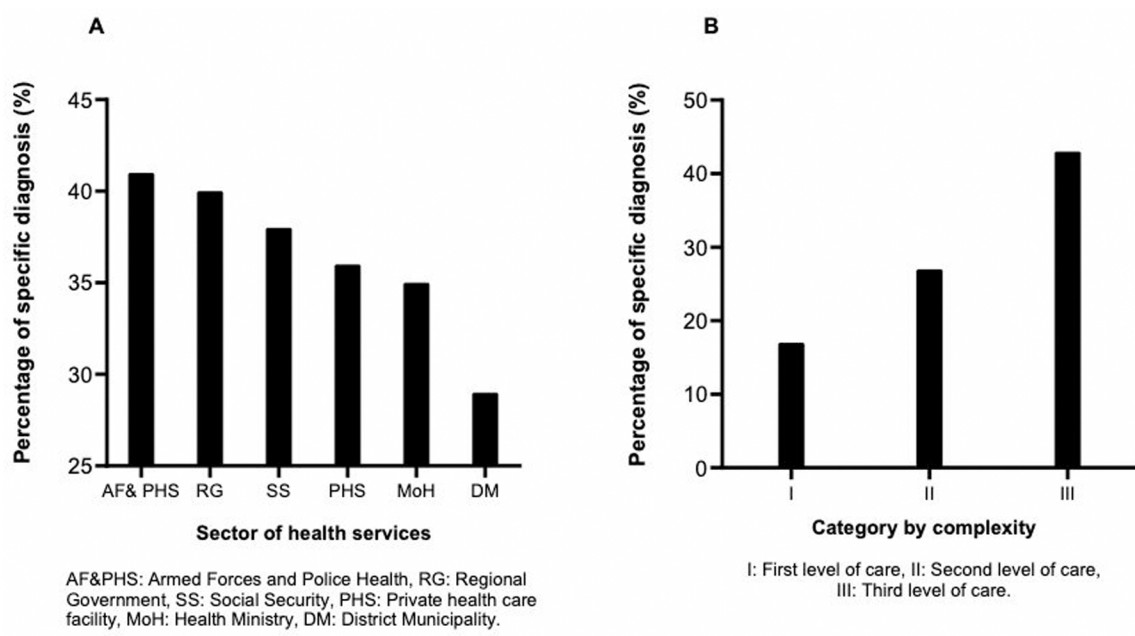

AF&PHS: Armed Forces and Police Health, RG: Regional Government, SS: Social Security, PHS: Private health care facility, MoH: Health Ministry, DM: District Municipality.

I: First level of care, II: Second level of care, III: Third level of care.

**Fig 6. Proportion of diagnoses specified according to health sector and IPRESS category.**

**Table 3. Most frequent diseases by subcategory and age group.**

| Age group (y) | Most frequent diseases by subcategory | | | | | | Total |
|---|---|---|---|---|---|---|---|
| | 1° | | 2° | | 3° | | |
| ‹ 1 | Intracerebral haemorrhage, intraventricular | 18,2% | Nontraumatic subdural haemorrhage | 12,3% | Intracerebral haemorrhage in hemisphere, unspecified | 5,9% | 424 |
| 1–9 | Intracerebral haemorrhage in hemisphere, unspecified | 12,3% | Nontraumatic subdural haemorrhage | 11,9% | Intracerebral haemorrhage in hemisphere, subcortical | 6,4% | 918 |
| 10–19 | Intracerebral haemorrhage in hemisphere, unspecified | 8,5% | Nontraumatic subdural haemorrhage | 6,4% | Intracerebral haemorrhage in hemisphere, subcortical | 4,6% | 1381 |
| 20–29 | Intracerebral haemorrhage in hemisphere, unspecified | 6,1% | Nontraumatic subdural haemorrhage | 5,2% | Intracerebral haemorrhage in hemisphere, subcortical | 3,9% | 2095 |
| 30–39 | Nontraumatic subdural haemorrhage | 4,4% | Intracerebral haemorrhage in hemisphere, unspecified | 4,4% | Intracerebral haemorrhage in hemisphere, subcortical | 3,9% | 3183 |
| 40–49 | Cerebral infarction due to embolism of cerebral arteries | 4,9% | Intracerebral haemorrhage in hemisphere, subcortical | 4,3% | Intracerebral haemorrhage in hemisphere, unspecified | 3,9% | 6239 |
| 50–59 | Cerebral infarction due to embolism of cerebral arteries | 5,2% | Intracerebral haemorrhage in hemisphere, subcortical | 4,0% | Intracerebral haemorrhage in hemisphere, unspecified | 3,2% | 11660 |
| 60+ | Cerebral infarction due to embolism of cerebral arteries | 5,9% | Cerebral infarction due to thrombosis of cerebral arteries | 4,2% | Nontraumatic subdural haemorrhage | 3,2% | 63876 |

The table shows the proportions of hospital discharges by decade.

Since most severe cases of stroke are treated in hospitals, we consider the observed hospital discharges to be a reasonable approximation of the clinical burden of stroke in Peru. However, we recognize that some mild cases may not have been captured in the database used, leading to potential underreporting, or that there may be inaccuracies in the discharge diagnoses recorded by healthcare facilities.

When comparing our results with other Latin American countries, we observed that Ecuador reported an incidence of 6.14 per 10,000 person-years for the 2015–2020 period [5]. In Matão, Brazil, a cumulative incidence of 10.8 per 10,000 persons was estimated for a first stroke event between 2003 and 2004 [26]. In Ñuble, Chile, incidence was significantly higher, with 22.94 per 10,000 persons during 2015–2016 [27]. In contrast, another study in Chile that only considered hospital discharges reported a lower incidence of 2.74 per 10,000 between 2018 and 2021 [4]. Our methodology is similar to that used in Ecuador and with the Chilean hospital discharge study, while the studies in Ñuble and Matão also included outpatient cases, which may have increased their incidence rates. Outpatient records could capture follow-ups of previously hospitalized stroke cases, as well as mild episodes that did not require hospitalization.

Our study found a higher incidence of stroke in men throughout the period analyzed, which aligns with global patterns [28]. This could be attributed to risk behaviors common in this population, such as higher alcohol [29,30] and tobacco consumption [31,32], both recognized risk factors for stroke. However, in 2019, mortality from stroke in the United States was higher in women than in men [33]. This may be related to prolonged estrogenic hormone exposure, which is a risk factor in women [34], given the global demographic shift towards an aging population.

We observed differences in the cumulative incidence of stroke across departments and throughout the study period. These variations reflect inequalities in healthcare access and service capacity, both common in countries with socioeconomic disparities. Differences in infrastructure, access to medical resources, and specialized equipment affect hospitals' ability to respond effectively [35]. In Lima and Callao, where most healthcare facilities are concentrated, hospital overcrowding may exacerbate these fluctuations, as hospitals in areas with structural limitations often cannot handle all cases or perform accurate diagnoses, impacting the reported incidence rates.

In 2020, the Pasco department presented an unusual stroke incidence of 14.7 cases per 10,000 persons, contrasting with previous and subsequent years, which recorded incidences of 0.6 and 0.5, respectively. This peak appears to be an

isolated case, likely related to errors in coding or database cleaning processes, which may have led to an overestimation of cases that year. Given this scenario, we recommend interpreting these results with caution, as they may not accurately reflect the true incidence of stroke in Pasco during 2020.

We observed a decrease in 2020 and 2021 (2.42 and 2.71 per 10,000 persons, respectively), coinciding with the COVID-19 pandemic. The health crisis redirected resources and hospital beds towards COVID-19 management, affecting the care of other diseases and the availability of medical personnel [36], with temporary facilities set up specifically for COVID-19 care [37]. Additionally, the high mortality rate, with 188,708 deaths, significantly impacted the country's overall mortality [38]. COVID-19 may have acted as a competing risk by reducing hospitalizations and death certificates for stroke, thereby altering the measured incidence. Thus, we suggest interpreting the 2020 and 2021 incidences with caution.

In our study, we observed that ischemic stroke was predominant across all age groups, while hemorrhagic forms, such as intracerebral and subdural hemorrhage, were more common in young people and children. This is consistent with other studies suggesting that the prevalence of arteriovenous malformations and drug use could be determining factors up to 39 years of age [39–41]. However, from age 40 onward, ischemic stroke predominates, likely due to risk factors such as hypertension, diabetes, atrial fibrillation, physical inactivity, and smoking [42]. Hypertension damages small arteries, increasing the risk of lacunar ischemia, while diabetes causes pathological changes in cerebral arteries, worsening prognosis in ischemic events, especially in those with poor glycemic control [43–47]. Atrial fibrillation causes blood stasis and favors the formation of cerebral thromboembolisms, increasing the risk of ischemic stroke [48].

Differential diagnosis between ischemic and hemorrhagic stroke relies mainly on medical history and physical examination [13]. We found that Armed Forces and National Police healthcare services and level III facilities had a higher proportion of specified diagnoses, although more than 50% remained unspecified. In high-complexity centers (categories II and III), most diagnoses should be specified. For example, studies using administrative records in Canada [49], Australia [50], and Spain [51] showed more than 80% of specified diagnoses, considering that these countries have specific programs for cerebrovascular disease. In category I centers, the low percentage of specified diagnoses is due to the lack of precise diagnostic tools and the rapid referral of patients to more complex centers, which is recorded as a hospital discharge.

The study has limitations. We used a secondary database of hospital discharges, which may lead to underreporting of cases due to errors in diagnostic coding or incorrect assignment of ICD-10 codes. However, inpatient discharge records are kept by staff with more training and experience, which could generate reliable data for estimating the incidence of this disease. In addition, SUSALUD prepares the database from the formal records of each IPRESS and, after validation of the information, and correction of the errors found. Another limitation is that there may be patients who did not manage to access health facilities due to death or mild clinical manifestations. Regarding asymptomatic cases or mild clinical manifestations, it has been reported that their prevalence varies from 8 to 28% [52–54]. In a study in the United States where there were 167,366 deaths from stroke, 47.6% of deaths occurred before transport and 0.7% were found dead on arrival of health personnel at the scene [55]. The clinical information in the database analyzed does not allow us to distinguish whether the case was a first episode of stroke or recurrent episodes. The classification of the 60 + age group limits detailed analysis, as it is not subdivided into decades. Without this disaggregation, the ability to identify age-specific patterns in older populations is reduced. Furthermore, we excluded category 0 from the complexity analysis, as it refers to institutions not yet categorized. Finally, incidence calculations were based on population projections, which may result in inaccurate data. However, it is important to note that demographic changes in the Peruvian population were not significant during this period.

Despite its limitations, this study provides evidence to estimate hospital bed demand due to stroke at both departmental and national levels, and highlights the need to prioritize actions that encourage preventive measures—such as lifestyle changes, detection, and control of risk factors—in regions with the highest incidence of this disease. It also highlights the need to improve database quality. This work can serve as a starting point for future studies, such as sensitivity analyzes

to assess the impact of uncaptured cases in hospital discharges. Additionally, further research is recommended to explore regional inequalities and access to healthcare services. Finally, an analysis of the impact of unspecified diagnoses on stroke data quality in countries with limited records is suggested, as this could be key to improving epidemiological surveillance.

## Conclusions

Our study demonstrates a heterogeneous incidence of cerebrovascular disease (stroke) across Peru, with elevated rates among males and individuals over 60. Cerebral infarction predominated as the primary diagnosis, especially in those over 40, while nearly half of the cases lacked specific diagnostic details. These results can guide the planning of health resources and interventions, with a comprehensive strategy that combines preventive, educational and diagnostic capacity-building measures in health services with a territorial approach. Lima is one of the departments that exceeds the national average over 7 years, and at the same time concentrates the greatest number of specialized professional resources and the hospital infrastructure with the greatest resolution capacity. This finding reflects the unequal distribution of resources in the health system to deal with cases of stroke. Based on our findings, we recommend enhancing diagnostic capacity in the country's emergency services by incorporating imaging tests, particularly in second-level complexity facilities and those outside the capital. Additionally, strengthening the referral system in primary care facilities is essential to ensure timely and appropriate case management. It is essential to improve the distribution of physicians trained in stroke management across healthcare facilities in Peru's regions, where one-third of these patients receive treatment. Finally, it is necessary to strengthen the response capacity of the health system during health emergencies, since the COVID-19 pandemic impacted the supply of hospital care for these patients at the national level.

## Supporting information

**S1 Table. Raw data.** *Deleted year and department.
(XLSX)

**S2 Table. Frequency of hospital discharges due to cerebrovascular disease by department.**
(DOCX)

**S3 Table. Specified diagnosis frequency by department.**
(DOCX)

## Author contributions

**Conceptualization:** Diego Maximiliano Guevara Rodríguez, Juan Diego Pichihua Grandez, Fabrizio Valdivia Dianderas.

**Data curation:** Diego Maximiliano Guevara Rodríguez, Juan Diego Pichihua Grandez, Fabrizio Valdivia Dianderas.

**Formal analysis:** Diego Maximiliano Guevara Rodríguez, Juan Diego Pichihua Grandez, Fabrizio Valdivia Dianderas, José Del Carmen Sara.

**Investigation:** Diego Maximiliano Guevara Rodríguez, Juan Diego Pichihua Grandez, Fabrizio Valdivia Dianderas, José Del Carmen Sara.

**Methodology:** Diego Maximiliano Guevara Rodríguez, Juan Diego Pichihua Grandez, Fabrizio Valdivia Dianderas, José Del Carmen Sara.

**Project administration:** Diego Maximiliano Guevara Rodríguez, Fabrizio Valdivia Dianderas.

**Resources:** José Del Carmen Sara.

**Software:** Diego Maximiliano Guevara Rodríguez, Juan Diego Pichihua Grandez, Fabrizio Valdivia Dianderas.

**Supervision:** Diego Maximiliano Guevara Rodríguez, Juan Diego Pichihua Grandez, Fabrizio Valdivia Dianderas, José Del Carmen Sara.

**Validation:** Diego Maximiliano Guevara Rodríguez, Juan Diego Pichihua Grandez, Fabrizio Valdivia Dianderas, José Del Carmen Sara.

**Visualization:** Diego Maximiliano Guevara Rodríguez, Juan Diego Pichihua Grandez, Fabrizio Valdivia Dianderas, José Del Carmen Sara.

**Writing – original draft:** Diego Maximiliano Guevara Rodríguez, Juan Diego Pichihua Grandez, Fabrizio Valdivia Dianderas, José Del Carmen Sara.

**Writing – review & editing:** Diego Maximiliano Guevara Rodríguez, Juan Diego Pichihua Grandez, Fabrizio Valdivia Dianderas, José Del Carmen Sara.

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
