## [Decision Letter · Decision Letter 0]

16 Jan 2025

PGPH-D-24-02632

Incidence of cerebrovascular disease in Peru from 2015 to 2023

Dear Dr. Guevara Rodríguez,

Thank you for submitting your manuscript to PLOS Global Public Health. After careful consideration, we feel that it has merit but does not fully meet PLOS Global Public Health’s publication criteria as it currently stands. Therefore, we invite you to submit a revised version of the manuscript that addresses the points raised during the review process.

We look forward to receiving your revised manuscript.

Kind regards,

Tarang Parekh

Academic Editor

Journal Requirements:

1. Please provide an Author Summary. This should appear in your manuscript between the Abstract (if applicable) and the Introduction, and should be 150–200 words long. The aim should be to make your findings accessible to a wide audience that includes both scientists and non-scientists. Sample summaries can be found on our website under Submission Guidelines:

https://journals.plos.org/globalpublichealth/s/submission-guidelines#loc-parts-of-a-submission.

2. Figure 3: please (a) provide a direct link to the base layer of the map (i.e., the country or region border shape) and ensure this is also included in the figure legend; and (b) provide a link to the terms of use / license information for the base layer image or shapefile. We cannot publish proprietary or copyrighted maps (e.g. Google Maps, Mapquest) and the terms of use for your map base layer must be compatible with our CC-BY 4.0 license. 

Additional Editor Comments (if provided):

The Editor has reviewed the manuscript alongside the reviewer comments. The reviewers generally agree that the manuscript presents an important descriptive update on Cerebrovascular disease from Peru , which is a very important and under-researched topic. However, they raised several points for clarification and improvement. Personally, I have issue with the abbreviation of CVD, which globally recognized as cardiovascular disease and not cerebrovascular disease. This gives a false impression to readers and I advise authors to avoid that. Further, the health establishment categories are confusing and missing information. The method to calculate incidence rate is not clear, and authors may consider providing equation to simplify using excel. Is there a reason to use word per 10,000 "inhabitants"? Please provide high resolution images.

Reviewers' comments:

Reviewer's Responses to Questions

**Comments to the Author**

1. Does this manuscript meet PLOS Global Public Health’s publication criteria ? Is the manuscript technically sound, and do the data support the conclusions? The manuscript must describe methodologically and ethically rigorous research with conclusions that are appropriately drawn based on the data presented.

Reviewer #1: Yes

Reviewer #2: Partly

2. Has the statistical analysis been performed appropriately and rigorously?

Reviewer #1: No

Reviewer #2: Yes

3. Have the authors made all data underlying the findings in their manuscript fully available (please refer to the Data Availability Statement at the start of the manuscript PDF file)?

Reviewer #1: No

Reviewer #2: Yes

4. Is the manuscript presented in an intelligible fashion and written in standard English?

Reviewer #1: No

Reviewer #2: Yes

5. Review Comments to the Author

Reviewer #1: Overall Evaluation and Recommendation (Revised)

Strengths:

The study provides a valuable contribution to understanding the incidence of cerebrovascular disease in Peru, a critical public health issue. However, the manuscript would benefit from explicitly distinguishing cerebrovascular disease (CVD) from cardiovascular disease, as "CVD" typically refers to cardiovascular disease in global literature. This distinction is important to make sure readers understand.

The use of national hospital discharge data, stratified by age, sex, region, and healthcare sector, offers a comprehensive view of the disease burden and helps identify regional disparities in healthcare access and outcomes.

The manuscript aligns well with PLOS Global Public Health's mission of addressing health inequities, particularly by highlighting regional and demographic differences in the burden of cerebrovascular disease.

Suggestions for Improvement:

To avoid confusion with cardiovascular disease, the authors should use the term cerebrovascular disease throughout the manuscript. Where CVD is used, it should be clearly defined as cerebrovascular disease at first mention, and the abbreviation should be avoided if possible. If it is necessary to use CVD, the authors should ensure it is consistently clarified in the text.

The section discussing CVD as the "second leading cause of cardiovascular mortality" should be revised to clarify that the term refers to cerebrovascular disease. A more apparent distinction between cardiovascular and cerebrovascular disease will reduce any ambiguity for readers.

More rigorous statistical methods, such as regression analysis, to account for potential confounders (e.g., socioeconomic status and lifestyle factors) would add depth to the analysis and strengthen the robustness of the findings.

A more detailed discussion of the limitations related to underreporting, coding errors, and missing data would enhance transparency and help contextualize the study's results.

The conclusions could be strengthened with more specific public health recommendations, particularly for improving the diagnosis and treatment of cerebrovascular disease in regions with higher incidence rates.

Recommendation: This manuscript has the potential to significantly contribute to global public health knowledge, particularly for low—and middle-income countries. However, to ensure clarity and consistency throughout the manuscript, the authors should properly revise terminology to distinguish cerebrovascular disease from cardiovascular disease. With these revisions, improvements in statistical rigor, and a more detailed exploration of policy implications, the manuscript is recommended for publication in PLOS Global Public Health.

Reviewer #2: Review Comments to the Authors:

1. The rationale as well as implication of the study is not explicitly stated in the Introduction section; such as, more explanation needed regarding the relationship between cerebrovascular disease (CVD) incidence and health resource planning. It is not also obvious from the introductory lines about how the real incidence of CVD will help mitigating the inequalities in accessing healthcare among the population of Peru.

2. The study, although acknowledged the limitation of not including non-hospitalization records such as outpatient cases and mild episodes; but did not elicit the rationale behind this limitation and how this challenge was minimized as those cases cannot be left behind to calculate the real incidence of CVD.

3. The Conclusion section seems like a repetition of the findings. Also, the concluding remark such as, the relationship of study findings with increased hospital beds is not properly explained throughout the paper.

6. PLOS authors have the option to publish the peer review history of their article (what does this mean? ). If published, this will include your full peer review and any attached files.

**Do you want your identity to be public for this peer review?** For information about this choice, including consent withdrawal, please see our Privacy Policy .

Reviewer #1: No

Reviewer #2: **Yes: ** ANINDYA DAS

---

## [Decision Letter · Decision Letter 1]

9 Apr 2025

Incidence of cerebrovascular disease in Peru from 2015 to 2023

PGPH-D-24-02632R1

Dear Guevara Rodríguez,

We are pleased to inform you that your manuscript 'Incidence of cerebrovascular disease in Peru from 2015 to 2023' has been provisionally accepted for publication in PLOS Global Public Health.

Best regards,

Tarang Parekh

Academic Editor

Reviewer Comments (if any, and for reference):

Reviewer's Responses to Questions

**Comments to the Author**

1. If the authors have adequately addressed your comments raised in a previous round of review and you feel that this manuscript is now acceptable for publication, you may indicate that here to bypass the “Comments to the Author” section, enter your conflict of interest statement in the “Confidential to Editor” section, and submit your "Accept" recommendation.

Reviewer #1: All comments have been addressed

Reviewer #2: All comments have been addressed

2. Does this manuscript meet PLOS Global Public Health’s publication criteria ? Is the manuscript technically sound, and do the data support the conclusions? The manuscript must describe methodologically and ethically rigorous research with conclusions that are appropriately drawn based on the data presented.

Reviewer #1: Yes

Reviewer #2: Yes

3. Has the statistical analysis been performed appropriately and rigorously?

Reviewer #1: Yes

Reviewer #2: Yes

4. Have the authors made all data underlying the findings in their manuscript fully available (please refer to the Data Availability Statement at the start of the manuscript PDF file)?

Reviewer #1: Yes

Reviewer #2: Yes

5. Is the manuscript presented in an intelligible fashion and written in standard English?

Reviewer #1: Yes

Reviewer #2: Yes

6. Review Comments to the Author

Reviewer #1: (No Response)

Reviewer #2: The manuscript is now well established in terms of rationale as well as implication of the study. In line with these, authors have properly elucidated how the challenges regarding data collection were minimized. Moreover, the concluding remarks are now more apparent to showcase the importance of study findings.

7. PLOS authors have the option to publish the peer review history of their article (what does this mean? ). If published, this will include your full peer review and any attached files.

**Do you want your identity to be public for this peer review?** For information about this choice, including consent withdrawal, please see our Privacy Policy .

Reviewer #1: No

Reviewer #2: **Yes: ** ANINDYA DAS
